# Impact of measured versus estimated glomerular filtration rate-based screening on living kidney donor characteristics: A study of multiple cohorts

Jessica van der Weijden[1], Marco van Londen[1], Joke I. Roodnat[2], Marcia L. Kho[2], Jacqueline van de Wetering[2], Heinrich Kloke[3], Ine M. M. Dooper[3], Stephan J. L. Bakker[1], Gerjan Navis[1], Ilja M. Nolte[4], Martin H. De Borst[1], Stefan P. Berger[1] *

1 Division of Nephrology, Department of Internal Medicine, University of Groningen, University Medical Center Groningen, Groningen, The Netherlands, 2 Division of Nephrology, Department of Internal Medicine, Erasmus MC, University Medical Center Rotterdam, Rotterdam, The Netherlands, 3 Department of Nephrology, Radboud University Medical Center, Nijmegen, The Netherlands, 4 Department of Epidemiology, University of Groningen, University Medical Center Groningen, Groningen, The Netherlands

* s.p.berger@umcg.nl

**Data Availability Statement:** Data cannot be shared publicly due to privacy regulations: the data contain patient-sensitive information and are

## Abstract

### Background

Most transplant centers in the Netherlands use estimated glomerular filtration rate (eGFR) for evaluation of potential living kidney donors. Whereas eGFR often underestimates GFR, especially in healthy donors, measured GFR (mGFR) allows more precise kidney function assessment, and therefore holds potential to increase the living donor pool. We hypothesized that mGFR-based donor screening leads to acceptance of donors with lower pre-donation eGFR than eGFR-based screening.

### Methods

In this longitudinal cohort study, we compared eGFR (CKD-EPI) before donation in one center using mGFR-based screening (mGFR-cohort, n = 250) with two centers using eGFR-based screening (eGFR-cohort1, n = 466 and eGFR-cohort2, n = 160). We also compared differences in eGFR at five years after donation.

### Results

Donor age was similar among the cohorts (mean±standard deviation (SD) mGFR-cohort 53±10 years, eGFR-cohort1 52±13 years, P = 0.16 vs. mGFR-cohort, and eGFR-cohort2 53±9 years, P = 0.61 vs. mGFR-cohort). Estimated GFR underestimated mGFR by 10±12 mL/min/1.73m$^2$ (mean±SD), with more underestimation in younger donors. In the overall cohorts, mean±SD pre-donation eGFR was lower in the mGFR-cohort (91±13 mL/min/1.73m$^2$) than in eGFR-cohort1 (93±15 mL/min/1.73m$^2$, P<0.05) and eGFR-cohort2 (94±12 mL/min/1.73m$^2$, P<0.05). However, these differences disappeared when focusing on more recent years, which can be explained by acceptance of more older donors with lower pre-

bound to national and European privacy regulations (General Data Protection Regulation, GDPR). Data can be made available to researchers who meet the criteria for access to confidential data through the UMCG Institutional Data Access / Ethics Committee (contact via RC@int.umcg.nl).

**Funding:** The authors received no specific funding for this work.

**Competing interests:** The authors have declared that no competing interests exist.

donation eGFR over time in both eGFR-cohorts. Five years post-donation, mean±SD eGFR was similar among the centers (mGFR-cohort 62±12 mL/min/1.73m$^2$, eGFR-cohort1 61±14 mL/min/1.73m$^2$, eGFR-cohort2 62±11 mL/min/1.73m$^2$, P = 0.76 and 0.95 vs. mGFR-cohort respectively). In the mGFR-cohort, 38 (22%) donors were excluded from donation due to insufficient mGFR with mean±SD mGFR of 71±9 mL/min/1.73m$^2$.

## Conclusions

Despite the known underestimation of mGFR by eGFR, we did not show that the routine use of mGFR in donor screening leads to inclusion of donors with a lower pre-donation eGFR. Therefore eGFR-based screening will be sufficient for the majority of the donors. Future studies should investigate whether there is a group (e.g. young donors with insufficient eGFR) that might benefit from confirmatory mGFR testing.

## Introduction

Living kidney donor transplantation currently represents ~50% of the total kidney transplantations in the Netherlands [1]. The main goal of living kidney donor evaluation is to assess whether a donor is healthy enough to undergo surgery and maintain good health after the nephrectomy [2, 3]. An important part of screening consists of estimation and/or measurement of the glomerular filtration rate (GFR) before donation to determine whether the donor will retain sufficient kidney function after donation for life long safe kidney function. Glomerular filtration rate can easily be estimated (eGFR) by various equations based on serum creatinine or cystatin C, but the gold standard is assessment of the GFR by measuring the clearance of exogenous filtration markers (mGFR) [4]. The latter is expensive and laborious and therefore much less widespread in use in the Netherlands [5].

There is no consensus regarding the best method for kidney function assessment during the selection of living donors [2, 3, 6]. Some guidelines advise eGFR based on the chronic kidney disease epidemiology collaboration (CKD-EPI) equation, others advise use of 24h creatinine clearance or even mGFR. Due to cost and time advantages, most centers in the Netherlands estimate GFR based on creatinine clearance. The University Medical Center Groningen is the only center in the Netherlands that routinely performs mGFR measurements in every (potential) donor.

Even though mGFR is considered the gold standard, it is unclear whether its use is advantageous over the use of eGFR for living kidney donor screening. A well-known limitation in white populations of kidney function estimation equations based on serum creatinine is that in the higher ranges of GFR, true GFR is underestimated [7–13]. Consequently, donors with normal to high kidney function might be mistakenly classified as having insufficient kidney function when eGFR is used, possibly leading to exclusion from donation. This study aimed to compare pre- and post-donation eGFR of living kidney donors between two centers that base the decision to accept a donor based on eGFR and a center that uses mGFR for decision making. We hypothesized that mGFR-based screening allows acceptance of donors with lower mean pre-donation eGFR compared to the population from centers that use eGFR-based screening. In addition, post-donation safety was studied by comparing kidney function five years after donation in donors who have been evaluated using mGFR and eGFR.

## Materials and methods

### Study design

In this longitudinal cohort study in the Netherlands, we compared effective living kidney donors between one center that used mGFR-based donor evaluation (University Medical Center Groningen, mGFR-cohort) and two centers that used eGFR-based donor evaluation (eGFR-cohort1 = Erasmus MC, University Medical Center Rotterdam, and eGFR-cohort2 = Radboud University Medical Center Nijmegen,). The study was approved by the institutional ethical review board of each participating center. For the mGFR-cohort, the study underwent ethical review in accordance with current ethical guidelines in 2014 as the TransplantLines biobank and cohort study (2014/077). The study was registered at clinicaltrials.gov under identifier NCT0327284 [14]. All donors included in the study signed informed consent for the use of their medical data for scientific research. In eGFR-cohort1 the study was approved by the EMC Medical Ethical Committee MEC-2019-0737. In eGFR-cohort1 and eGFR-cohort2, all donors have given written informed consent for the use of their medical data for scientific research. All procedures were conducted in accordance with the Declaration of Helsinki, Declaration of Istanbul, and the Dutch Scientific Guidelines.

### Study population and measurements mGFR-cohort

In the University Medical Center Groningen, the selection criteria according to Dutch Living Kidney Donor guidelines (based on international guidelines) were used [3]. Instead of the recommended eGFR, mGFR was used to assess renal function before and after donation. A total of 1,113 potential living kidney donors were screened between 2006 and 2018 in Groningen. In this group, 977 donors were accepted for donation, of which 250 donated and had data for five-year follow-up available. The mGFR, measured as the urinary clearance of $^{125}$I-iothalamate (S1 File), and eGFR (based on serum creatinine) were measured before donation in every (potential) donor and five years after donation. Measured GFR was corrected for body surface area (BSA, calculated according to Dubois et al.) [15]. Clinical decision making was based on pre-donation mGFR. Estimated GFR was retrospectively determined according to the Chronic Kidney Disease Epidemiology Collaboration (CKD-EPI) equation to enable comparison with the eGFR-cohorts and according to the Modification of Diet in Renal Disease (MDRD) equation and the Cockcroft-Gault (CG) equation for secondary analyses [12, 16, 17]. Twenty-four-hour urine samples were used to calculate the 24h creatinine clearance (CrCl). Besides kidney function measurements, clinical parameters such as weight, height, and blood pressure were measured during the visits. Blood pressure was measured three times while seated with an interval of three minutes and a fourth time after standing straight for one minute using an automatic device as described previously [14].

### Study population and measurements eGFR-cohort1

Between 1981 and 2019, 4,801 potential donors were screened for donation at the Erasmus MC, University Medical Center Rotterdam, the Netherlands. Of these donors, 2,144 donors eventually donated. For 647 donors, five-year follow-up was available. In order to enable comparison with the mGFR-cohort, donors that were screened before 2006 were excluded, rendering 466 donors eligible for this study. Glomerular filtration rate was assessed by equations based on serum creatinine, measured by enzymatic creatinine determination. In potential donors with unexpectedly low eGFR, 24-hour urine collection was performed to calculate endogenous creatinine clearance. When CrCl was adequate donation was allowed. Besides

kidney function measurements, clinical parameters such as weight, height and blood pressure were measured during the visits.

## Study population and measurements eGFR-cohort2

Between 2006 and 2014, 970 potential donors were screened for donation in the Radboud University Medical Center in Nijmegen, the Netherlands. Of these donors, 603 donors donated in these years. For 160 donors, five-year follow-up was available. Glomerular filtration rate was assessed by the equations based on serum creatinine and two 24-hour urine collection allowing calculation of the endogenous creatinine clearance. Serum creatinine was measured by enzymatic method. Besides kidney function measurements, clinical parameters such as weight, height and blood pressure were measured during the visits. The office blood pressure measurement was included in this study.

## Statistical analyses

Data are presented as mean±standard deviation (SD) for normally distributed variables and as median (first quartile–third quartile) for non-normally distributed variables. The distribution was tested using histograms and probability plots. Binary variables are shown as 'number (%)'. Measured GFR data are reported as absolute values (mL/min) and corrected for body surface area according to Dubois et al. (mL/min/1.73m$^2$) [15]. To maintain consistency and enable comparison, eGFR was recalculated according to the CKD-EPI equation for all centers. Differences in characteristics of donors between the mGFR-cohort and eGFR-cohort1 and between the mGFR-cohort and eGFR-cohort2 were tested using the independent Student's t-test for normally distributed variables, the Mann-Whitney U-test for non-normally distributed variables, and the chi-square test for proportions. To characterize donors with low pre-donation eGFR, we compared characteristics of 10% of donors with the lowest pre-donation eGFR to the other 90% of the donors using the tests mentioned above. Similarly, we compared donors with an underestimation of mGFR $\geq$10 and $\geq$20 mL/min/1.73m$^2$ by eGFR to donors with no underestimation or an underestimation <10 and <20 mL/min/1.73m$^2$, in order to identify donors at risk of underestimation by eGFR. Bias between pre- and post-donation eGFR and mGFR was calculated as the mean difference between both parameters. Because reason of exclusion from donation was mostly multifactorial and rarely solely dependent GFR, we did not analyze the number of donors excluded based on kidney function per center. SPSS version 23 for Windows (IBM, Armonk, NY) and Graphpad Prism 8 for Windows (Graphpad, San Diego, CA) were used to perform the analyses. P values <0.05 were considered statistically significant.

## Results

### Bias between eGFR and mGFR

The known underestimation of pre-donation mGFR by pre-donation eGFR (CKD-EPI) was also present in the mGFR-cohort (mean±SD bias = -10±12 mL/min/1.73m$^2$, S1 Table). This underestimation was visualized in a Bland-Altman plot (Fig 1). This bias became smaller five years after donation (-5±9 mL/min/1.73m$^2$). Pre-donation 24h CrCl overestimated pre-donation mGFR with a bias of 26±29 mL/min (S2 Table). Five years after donation, this overestimation was still present, although it was slightly reduced (18±19 mL/min).

### Donors in whom pre-donation GFR was underestimated

The mGFR-cohort of donors was split into a group in which eGFR underestimated mGFR ($\geq$10 mL/min/1.73m$^2$ difference) and a group in which eGFR did not underestimate mGFR

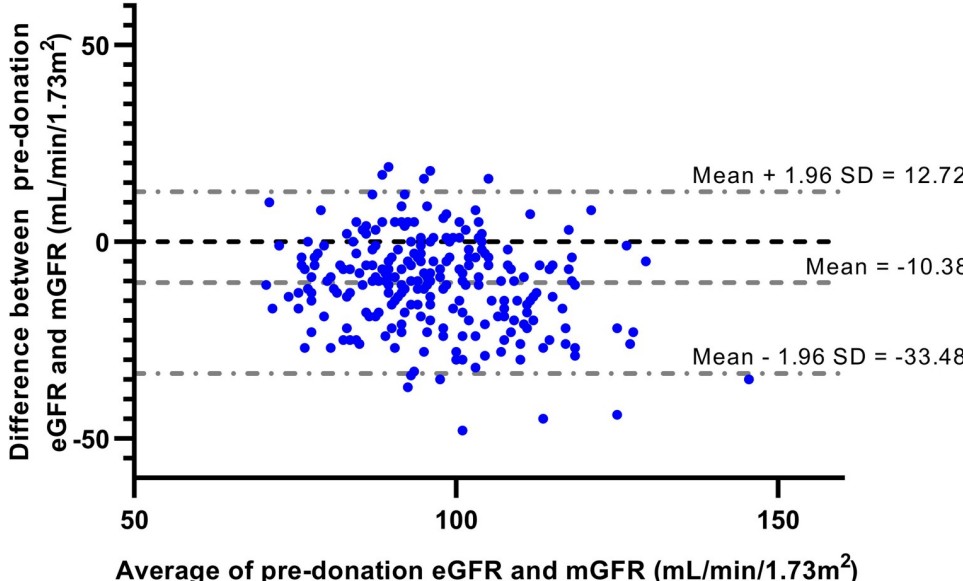

**Fig 1. Bland-altman plot of pre-donation eGFR and pre-donation mGFR.** Bias between pre-donation eGFR and pre-donation mGFR is shown on the X-axis, the average between pre-donation eGFR and pre-donation mGFR is shown on the Y-axis. Mean±SD bias was -10.38 mL/min/1.73m², the 95% confidence interval of the mean bias was -33.48 to 12.72 mL/min/1.73m².

(<10 mL/min/1.73m² difference), as shown in Table 1. Besides differences in kidney function, there were no statistically significant differences in clinical characteristics between donors in whom mGFR was underestimated by eGFR and donors in whom mGFR was not underestimated by eGFR. Donors in whom eGFR underestimated mGFR ≥20 mL/min/1.73m² were significantly younger than donors in whom the difference between eGFR and mGFR was <20 mL/min/1.73m² (mean±SD 50±8 vs. 54±10 years respectively, P = 0.02). A low eGFR compared to 24h CrCl was mainly limited to donors with higher height, weight, BMI and BSA (S3 Table). Difference between eGFR and 24h CrCl was more commonly <10 mL/min in female donors (S3 Table). An overestimation of mGFR by eGFR was present in 45 donors (S4 Table).

## Comparison of kidney function and clinical characterestics before donation

The characteristics of the living kidney donor populations before donation are shown in Table 2. Mean±SD age before donation was 53±10 (mGFR-cohort (Groningen)), 53±12 (eGFR-cohort1 (Rotterdam)), and 54±10 (eGFR-cohort2 (Nijmegen)) years and 54%, 54%, and 45%, respectively were female. Mean±SD eGFR (CKD-EPI) before donation was 91±13 mL/min/1.73m² in the mGFR-cohort, which was lower than in eGFR-cohort1 (93±15 mL/min/1.73m², P = 0.20) and eGFR-cohort2 (94±12 mL/min/1.73m², P = 0.01) where eGFR formed the basis for screening. Distributions of pre-donation eGFR (CKD-EPI) for the different centers are shown in Fig 2. Mean±SD mGFR$_{/BSA}$ before donation was 101±15 mL/min/1.73m² in the mGFR-cohort. Pre-donation systolic blood pressure (SBP) was higher in eGFR-cohort2 (137±16 mmHG) compared to the mGFR-cohort (128±14 mmHg, P<0.001) and slightly different between eGFR-cohort1 (130±16 mmHg) and the mGFR-cohort (P = 0.05). This difference is probably explained by the use of office blood pressure in eGFR-cohort2. Body size measurements (height, weight, BMI and BSA) did not show major differences before and after donation between the cohorts.

**Table 1. Pre-donation characteristics of donors from the mGFR-cohort with an underestimation of $mGFR_{BSA}$ by eGFR $\geq$10 mL/min/1.73m$^2$.**

| | *Underestimation $\geq$10 mL/min/1.73m$^2$* | *Underestimation <10 mL/min/1.73m$^2$* | **P value** |
|---|---|---|---|
| Number, n (%) | 121 (49) | 127 (51) | - |
| CKD-EPI, mL/min/1.73m$^2$ | 88 ±13 | 94 ±12 | <0.001 |
| CrCl, mL/min | 131 ±32 | 124 ±35 | <0.001 |
| mGFR, mL/min | 122 ±24 | 109 ±19 | <0.001 |
| mGFR$_{/BSA}$, mL/min/1.73m$^2$ | 108 ±16 | 96 ±12 | <0.001 |
| Age, years | 52 ±9 | 53 ±10 | 0.31 |
| Sex, n (%) female | 61 (50) | 72 (57) | 0.32 |
| Race, n (%) Caucasian | 121 (100) | 127 (100) | - |
| Weight, kg | 80 ±13 | 81 ±14 | 0.55 |
| Height, cm | 174 ±9 | 174 ±9 | 0.98 |
| BMI, kg/m$^2$ | 26 ±3 | 27 ±4 | 0.38 |
| BSA, m$^2$ | 1.95 ±0.20 | 1.96 ±0.20 | 0.70 |
| SBP, mmHg | 128 ±14 | 127 ±14 | 0.45 |
| DBP, mmHg | 77 ±9 | 76 ±9 | 0.48 |
| Serum creat, µmol/L | 78 ±13 | 70 ±12 | <0.001 |
| | *Underestimation $\geq$20 mL/min/1.73m$^2$* | *Underestimation <20 mL/min/1.73m$^2$* | **P value** |
| Number, n (%) | 53 (21) | 195 (79) | - |
| CKD-EPI, mL/min/1.73m$^2$ | 89 ±14 | 92 ±12 | 0.20 |
| CrCl, mL/min | 139 ±28 | 124 ±34 | 0.01 |
| mGFR, mL/min | 133 ±23 | 110 ±20 | <0.001 |
| mGFR$_{/BSA}$, mL/min/1.73m$^2$ | 116 ±15 | 98 ±12 | <0.001 |
| Age, years | 50 ±8 | 54 ±10 | 0.02 |
| Sex, n (%) female | 23 (43) | 110 (56) | 0.09 |
| Race, n (%) Caucasian | 195 (100) | 53 (100) | - |
| Weight, kg | 82 ±13 | 81 ±14 | 0.46 |
| Height, cm | 176 ±10 | 174 ±9 | 0.28 |
| BMI, kg/m$^2$ | 26 ±3 | 26 ±4 | 0.99 |
| BSA, m$^2$ | 1.98 ±0.20 | 1.95 ±0.20 | 0.32 |
| SBP, mmHg | 127 ±14 | 128 ±14 | 0.62 |
| DBP, mmHg | 76 ±9 | 77 ±9 | 0.43 |
| Serum creat, µmol/L | 79 ±15 | 73 ±12 | <0.001 |

Binary variables presented as n (%), continuous variables presented as mean ±SD

Abbreviations: CKD-EPI: Chronic kidney disease epidemiology collaboration equation; CrCl: Creatinine clearance; mGFR: Measured GFR; BMI: Body mass index; BSA: Body surface area; SBP: Systolic blood pressure; DBP: Diastolic blood pressure; SD: Standard deviation.

## Analysis of differences in eGFR over time

Because this study included donors that were screened during a large timeframe (especially in eGFR-cohort1), we performed secondary analyses to investigate whether the differences in pre-donation eGFR were consistent over time. We therefore split the cohort in two equal parts, which resulted in a group that was screened before 01-01-2009, and a group that was screened after 01-01-2009. Fig 3 shows the distribution of pre-donation eGFR (CKD-EPI) before and after 2009 and shows that the differences that were seen in the total cohort, mainly depended on differences in pre-donation eGFR before 2009 (mean±SD eGFR in mGFR-cohort: 90±12 mL/min/1.73m$^2$, eGFR-cohort1: 94±15 mL/min/1.73m$^2$, eGFR-cohort2: 97±11 mL/min/1.73m$^2$). When focusing on data after 2009, the differences in pre-donation eGFR seem to disappear (mGFR-cohort: 92±13 mL/min/1.73m$^2$, eGFR-cohort1: 92±15 mL/min/

**Table 2. Characteristics of the living kidney donors during screening.**

| | mGFR-cohort | eGFR-cohort1 | P vs. mGFR-cohort | eGFR-cohort2 | P vs. mGFR-cohort |
|---|---|---|---|---|---|
| Number, n (%) | 250 | 466 | - | 160 | - |
| CKD-EPI, mL/min/1.73m$^2$ | 91 ±13 | 93 ±15 | 0.20 | 94 ±12 | **0.02** |
| CrCl, mL/min | 127 ±33 | - | - | 129 ±28 | 0.50 |
| mGFR, mL/min | 115 ±22 | - | - | - | - |
| mGFR$_{/BSA}$, mL/min/1.73m$^2$ | 101 ±15 | - | - | - | - |
| Age, years | 53 ±10 | 53 ±12 | 0.91 | 54 ±10 | 0.41 |
| Female sex, n (%) | 134 (54) | 252 (54) | 0.90 | 72 (45) | 0.89 |
| Caucasian race, n (%) | 250 (100) | 450 (97) | - | 160 (100) | - |
| Weight, kg | 80 ±14 | 79 ±14 | 0.12 | 78 ±14 | **0.05** |
| Height, cm | 174 ±9 | 172 ±9 | **<0.001** | 173 ±8 | 0.13 |
| BMI, kg/m$^2$ | 26 ±3 | 27 ±4 | 0.38 | 26 ±4 | 0.13 |
| BSA, m$^2$ | 1.96 ±0.20 | 1.92 ±0.20 | **0.01** | 1.92 ±0.19 | 0.04 |
| SBP, mmHg | 128 ±14 | 130 ±16 | **0.05** | 137 ±16 | **<0.001** |
| DBP, mmHg | 76 ±9 | 78 ±9 | 0.07 | 81 ±8 | **<0.001** |
| Use of antihypertensive medication, n (%) | 43 (17) | 79 (17) | 0.93 | 24 (15) | 0.56 |
| Smoking, n (%) | 59 (24) | - | - | 52 (33) | **0.05** |
| Serum creat, μmol/L | 74 ±13 | 74 ±14 | 0.62 | 72 ±12 | 0.18 |

Binary variables presented as n (%), continuous variables presented as mean ±SD

Abbreviations: CKD-EPI: Chronic kidney disease epidemiology collaboration equation; CrCl: Creatinine clearance; mGFR: Measured GFR; BMI: Body mass index; BSA: Body surface area; SBP: Systolic blood pressure; DBP: Diastolic blood pressure; SD: Standard deviation.

1.73m$^2$, eGFR-cohort2: 93±12 mL/min/1.73m$^2$). When looking at age before and after 2009 (Fig 4), our data show that both eGFR-cohort1 and eGFR-cohort2 accepted older donors after 2009 compared to before 2009, although only significant in eGFR-cohort2 (mean±SD age eGFR-cohort1: 52±12 years before and 53±13 years after 2009 (P = 0.16), eGFR-cohort2: 51 ±10 before and 55±9 after 2009 (P = 0.01)), whereas in mGFR-cohort there does not seem to be a difference in age over time (before 2009 53±9 years vs. after 2009 53±10 years, P = 0.99). Mean BMI did not differ before and after 2009 in the three centers (S1 Fig).

### Living kidney donor characteristics five years after donation

Five years after donation, there was no difference in mean±SD eGFR (CKD-EPI) in the total cohort (mGFR-cohort: 62±12 mL/min/1.73m$^2$, eGFR-cohort1: 60±14 mL/min/1.73m$^2$ (P = 0.15 vs. mGFR-cohort), eGFR-cohort2: 61±11 mL/min/1.73m$^2$ (P = 0.65 vs. mGFR-cohort) Table 3 and S2 Fig). When looking at differences between the centers for the groups that were screened before 2009 and after 2009, we see no differences between the centers, but for all centers five-year post-donation eGFR was lower (mGFR-cohort: 64±12 mL/min/1.73m$^2$ before and 60±12 mL/min/1.73m$^2$ after 2009 (P = 0.01), eGFR-cohort1: 61±14 mL/min/ 1.73m$^2$ before and 59±13 mL/min/1.73m$^2$ after 2009 (P = 0.07), eGFR-cohort2: 63±11 mL/ min/1.73m$^2$ before and 60±11 mL/min/1.73m$^2$ after 2009 (P = 0.04), S3 Fig).

### Secondary analyses of pre-donation kidney function

Mean±SD pre-donation 24-hour creatinine clearance (24h CrCl) was 127±33 mL/min in the mGFR-cohort and 129±28mL/min in eGFR-cohort2 (P = 0.50); eGFR-cohort1 did not routinely determine CrCl (S4 Fig). These results were similar before 2009 compared to after 2009.

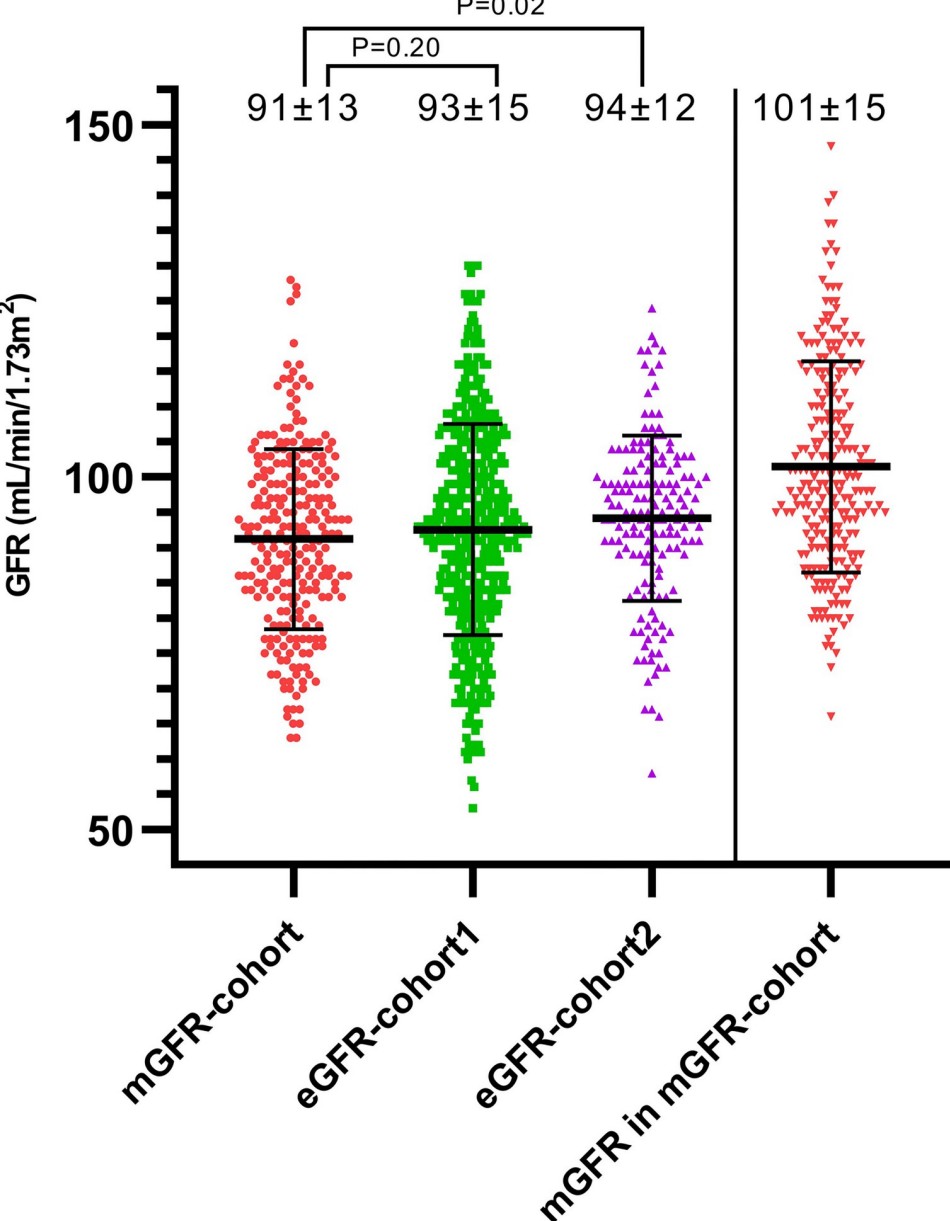

**Fig 2. Distribution of pre-donation eGFR (CKD-EPI) per center.** Differences between mean pre-donation eGFR were tested using the independent sample T-test, P-values are shown in the Fig. Distribution of mGFR in the mGFR-cohort was added on the right in the Fig.

We also compared pre-donation eGFR according to the CG and MDRD equation before and after 2009, which yielded similar results to the CKD-EPI comparison (S5 and S6 Figs).

## Comparison of donors with marginal pre-donation eGFR

We subsequently focused on the 10% of donors with lowest pre-donation eGFR in the three cohorts (Table 4). In these donors from the mGFR-cohort, mean±SD pre-donation

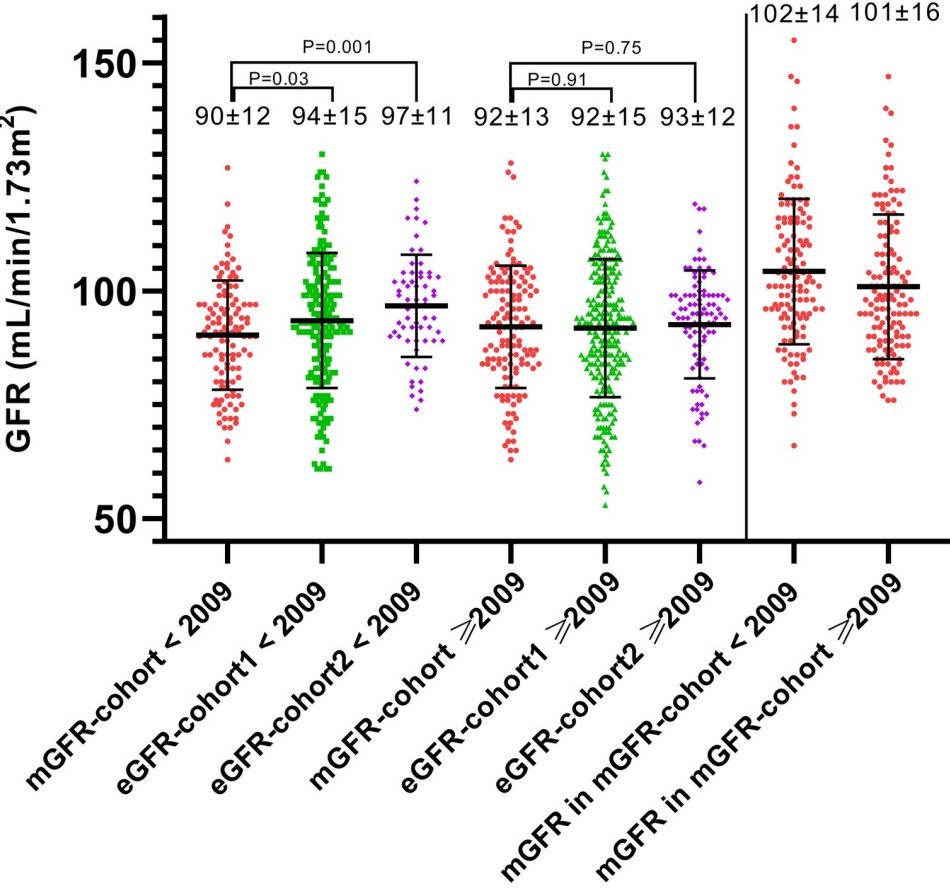

**Fig 3. Distribution of pre-donation eGFR (CKD-EPI) before and after 2009 per center.** Differences between mean pre-donation eGFR were tested using the independent sample T-test, P-values are shown in the Fig. Distribution of mGFR in the mGFR-cohort was added on the right in the Fig.

eGFR was 70±3 mL/min/1.73m$^2$ and mean±SD five-year post-donation eGFR was 48±6 mL/min/1.73m$^2$ (Table 4). Pre-donation mGFR$_{/BSA}$ was 86±9 mL/min/1.73m$^2$ and only decreased to 59±9 mL/min/1.73m$^2$ five years after donation. The 10% donors from eGFR-cohort1 and eGFR-cohort2 with lowest pre-donation eGFR were older than the corresponding donors from the mGFR-cohort (65±9 years and 60±8 years respectively vs. 56±6 years (P<0.001 and P = 0.09 respectively)). Furthermore, BSA tended to be higher in these donors from the mGFR-cohort versus eGFR-cohort1 and eGFR-cohort2 (1.94±0.19 m$^2$ vs. 1.89±0.15 m$^2$ and 1.89±0.17 m$^2$ (P = 0.13 and P = 0.30, respectively), but power might be too limited to draw conclusions. The same applies to blood pressure (132±21 mmHg for the mGFR-cohort vs. 136±17 mmHg for eGFR-cohort1 and 138±22 mmHg for eGFR-cohort2 (P = 0.34 and P = 0.52, respectively). In the mGFR-cohort, 5% of the donors had a pre-donation eGFR below the age-adapted threshold versus 3% in eGFR-cohort1 (P = 0.13) and 1% in eGFR-cohort2 (P = 0.04) (S5 Table). None of these donors had poor outcomes at five years after donation.

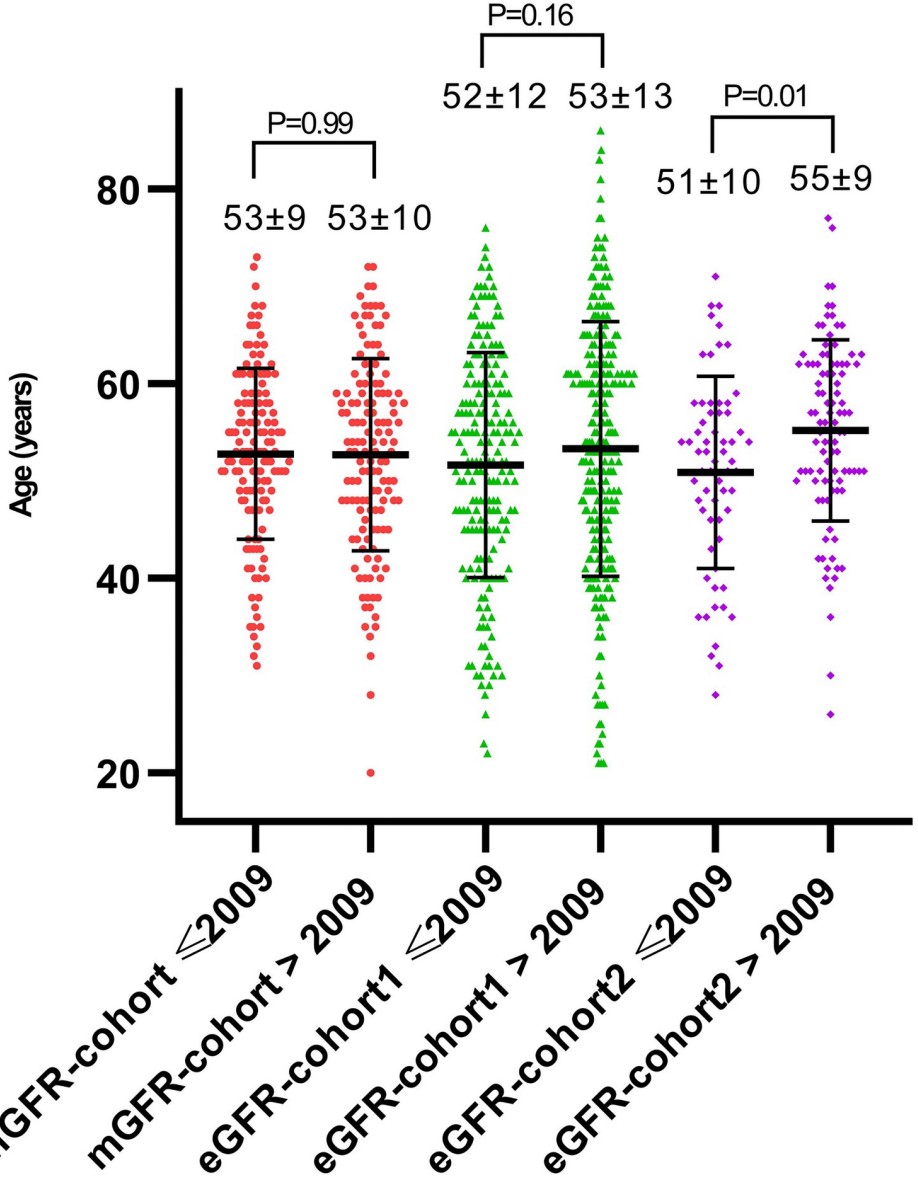

**Fig 4. Distribution of age before and after 2009 per center.** Differences between mean age were tested using the independent sample T-test, P-values are shown in the Fig.

### Donors that were excluded from donation in the mGFR-cohort

From 2006 to 2018, 173 potential donors were excluded from donation (Table 5). Mean±SD eGFR of these donors was 81±14 mL/min/1.73m$^2$ compared to 91±13 mL/min/1.73m$^2$ in the accepted group (P<0.001). In 16 of these donors, insufficient mGFR was the main reason for disapproval. In 20 donors, insufficient mGFR was one of multiple reasons for disapproval. In two donors, mGFR was considered too low for the recipient. The characteristics of the donors that were declined due to insufficient mGFR (N = 38) are also shown in Table 5. Mean±SD mGFR of these donors was 70±12 mL/min/1.73m$^2$ (P<0.001 vs. accepted donors). Female

**Table 3. Characteristics of the living kidney donors five years after donation.**

| | mGFR-cohort | eGFR-cohort1 | P vs. mGFR-cohort | eGFR-cohort2 | P vs. mGFR-cohort |
|---|---|---|---|---|---|
| Number, n | 250 | 466 | - | 160 | - |
| CKD-EPI, mL/min/1.73m$^2$ | 62 ±12 | 60 ±14 | 0.15 | 61 ±11 | 0.65 |
| ΔCKD-EPI, mL/min/1.73m$^{2*}$ | -29 ±10 | -32 ±10 | **<0.001** | -33 ±8 | **<0.001** |
| CrCl, mL/min | 85 ±22 | - | - | - | - |
| mGFR, mL/min | 76 ±16 | - | - | - | - |
| mGFR$_{/BSA}$, mL/min/1.73m$^2$ | 67 ±11 | - | - | - | - |
| Age, years | 58 ±10 | 58 ±12 | 0.74 | 59 ±10 | 0.29 |
| Weight, kg | 83 ±15 | 81 ±15 | 0.17 | 80 ±16 | 0.15 |
| BMI, kg/m$^2$ | 27 ±4 | 27 ±4 | 0.34 | 27 ±4 | 0.36 |
| BSA, m$^2$ | 1.98 ±0.21 | 1.94 ±0.20 | **0.02** | 1.94 ±0.21 | 0.10 |
| SBP, mmHg | 127 ±14 | 133 ±16 | **<0.001** | 133 ±15 | **<0.001** |
| DBP, mmHg | 76 ±10 | 79 ±9 | **<0.001** | 79 ±7 | **0.01** |
| Use of antihypertensive medication, n (%) | 67 (27) | 141 (30) | 0.33 | 58 (36) | **0.04** |
| Smoking, n (%) | 69 (28) | - | - | 46 (29) | 0.80 |
| Serum creat, μmol/L | 103 ±20 | 106 ±21 | 0.09 | 104±18 | 0.48 |

Binary variables presented as n (%), continuous variables presented as mean ±SD

*Calculated as: CKD-EPI 5 years after donation minus pre-donation CKD-EPI

Abbreviations: CKD-EPI: Chronic kidney disease epidemiology collaboration equation; CrCl: Creatinine clearance; mGFR: Measured GFR; BMI: Body mass index; BSA: Body surface area; SBP: Systolic blood pressure; DBP: Diastolic blood pressure; SD: Standard deviation.

donors were more likely to be declined fordonation due to low GFR (84% female in the "declined due to GFR" group vs.54% female in the accepted group (P<0.001)). Declined donors were also significantly older with smaller body size measurements compared to accepted donors.

**Table 4. Pre- and 5 year post-donation characteristics of 10% of the donors with lowest pre-donation eGFR per center.**

| | mGFR-cohort | | eGFR-cohort1 | | eGFR-cohort2 | |
|---|---|---|---|---|---|---|
| | *Pre-donation* | *5 year post-donation* | *Pre-donation* | *5 year post-donation* | *Pre-donation* | *5 year post-donation* |
| Number, n (%) | 25 | 25 | 51 | 51 | 16 | 16 |
| CKD-EPI, mL/min/1.73m$^2$ | 70 ±3 | 48 ±6 | 67 ±5 | 43 ±8 | 72 ±5 | 48 ±6 |
| CrCl, mL/min | 107 ±20 | 73 ±17 | - | - | 106 ±22 | - |
| mGFR, mL/min | 98 ±15 | 67 ±15 | - | - | - | - |
| mGFR$_{/BSA}$, mL/min/1.73m$^2$ | 87 ±9 | 59 ±9 | - | - | - | - |
| Age, years | 56 ±6 | 62 ±7 | 65 ±9 | 71 ±9 | 60 ±8 | 66 ±8 |
| Female sex, n (%) | 15 (60) | 15 (60) | 29 (57) | 29 (57) | 9 (56) | 9 (56) |
| Caucasian race, n (%) | 25 (100) | 25 (100) | 51 (100) | 51 (100) | 16 (100) | 16 (100) |
| Weight, kg | 80 ±12 | 82 ±13 | 77 ±9 | 79 ±11 | 76 ±12 | 79 ±14 |
| Height, cm | 173 ±9 | 173 ±9 | 171 ±8 | 171 ±8 | 172 ±9 | 172 ±9 |
| BMI, kg/m$^2$ | 27 ±3 | 27 ±3 | 26 ±3 | 27 ±3 | 26 ±3 | 27 ±3 |
| BSA, m$^2$ | 1.94 ±0.19 | 1.97 ±0.19 | 1.89 ±0.15 | 1.90 ±0.16 | 1.89 ±0.17 | 1.91 ±0.22 |
| SBP, mmHg | 132 ±21 | 130 ±18 | 136 ±17 | 135 ±16 | 138 ±22 | 131 ±10 |
| DBP, mmHg | 79 ±9 | 77 ±13 | 79 ±8 | 77 ±10 | 81 ±7 | 78 ±5 |
| Serum creat, μmol/L | 90 ±12 | 121 ±21 | 90 ±12 | 127 ±22 | 87 ±11 | 119 ±18 |

Binary variables presented as n (%), continuous variables presented as mean ±SD

Abbreviations: CKD-EPI: Chronic kidney disease epidemiology collaboration equation; CrCl: Creatinine clearance; mGFR: Measured GFR; BMI: Body mass index; BSA: Body surface area; SBP: Systolic blood pressure; DBP: Diastolic blood pressure.

**Table 5. Characteristics of "accepted", "declined" and "declined due to low mGFR" donors in the mGFR-cohort.**

| | Accepted[*] | Declined | P vs. accepted | Declined due to mGFR | P vs. accepted |
|---|---|---|---|---|---|
| Number, n | 250 | 173 | - | 38 | - |
| CKD-EPI, mL/min/1.73m$^2$ | 91 ±13 | 81 ±14 | <**0.001** | 70 ±12 | <**0.001** |
| CrCl, mL/min | 127 ±33 | 106 ±31 | <**0.001** | 77 ±23 | <**0.001** |
| mGFR, mL/min | 115 ±22 | 96 ±22 | <**0.001** | 72 ±8 | <**0.001** |
| mGFR$_{/BSA}$, mL/min/1.73m$^2$ | 101 ±15 | 88 ±18 | <**0.001** | 71 ±9 | <**0.001** |
| Age, years | 53 ±10 | 60 ±11 | <**0.001** | 66 ±6 | <**0.001** |
| Female sex, n (%) | 134 (54) | 100 (8) | 0.39 | 32 (84) | <**0.001** |
| Caucasian race, n (%) | 250 (100) | 173 (100) | - | 38 (100) | - |
| Weight, kg | 80 ±14 | 78 ±14 | 0.06 | 69 ±9 | <**0.001** |
| Height, cm | 174 ±9 | 171 ±9 | **0.001** | 167 ±6 | <**0.001** |
| BMI, kg/m$^2$ | 26 ±3 | 27 ±4 | 0.73 | 25 ±3 | **0.01** |
| BSA, m$^2$ | 1.96 ±0.20 | 1.90 ±0.20 | **0.01** | 1.77 ±12 | <**0.001** |
| SBP, mmHg | 128 ±14 | 131 ±14 | **0.02** | 129 ±10 | 0.50 |
| DBP, mmHg | 76 ±9 | 77 ±10 | 0.90 | 76 ±9 | 0.89 |
| Use of antihypertensive medication, n (%) | 43 (17) | 46 (27) | **0.02** | 11 (29) | 0.08 |
| Serum creat, μmol/L | 74 ±13 | 79 ±13 | <**0.001** | 82 ±15 | **0.002** |

[*]Donors who were accepted, donated and had 5-year follow-up available

Binary variables presented as n (%), continuous variables presented as mean ±SD

Abbreviations: CKD-EPI: Chronic kidney disease epidemiology collaboration equation; CrCl: Creatinine clearance; mGFR: Measured GFR; BMI: Body mass index; BSA: Body surface area; SBP: Systolic blood pressure; DBP: Diastolic blood pressure; SD: Standard deviation.

## Discussion

This study aimed to compare pre- and post-donation eGFR of living kidney donors between two centers that base the decision to accept a donor based on eGFR and a center that uses mGFR for decision making. We hypothesized that, due to systematic underestimation of mGFR by eGFR, mGFR-based screening allows acceptance of donors with lower pre-donation eGFR than a center that only uses eGFR. Findings confirm that pre-donation eGFR can indeed underestimate pre-donation mGFR, especially in younger donors. In the overall cohort, we found lower pre-donation eGFR in a center that uses mGFR for donor screening than in centers that use eGFR. However, when focusing on more recent data, these differences disappear, and therefore, routine use of mGFR for living kidney donor screening does not seem to add value compared to using eGFR on population level. Lastly, we did not find differences in five-year post-donation eGFR between centers that use eGFR- or mGFR-based donor screening.

Measuring the clearance of exogenous filtration markers is the best available method to assess GFR [18]. Because mGFR has cost and availability issues, eGFR equations are most widely used. In line with the literature, our results show an underestimation of mGFR by eGFR [7–13]. Mean pre-donation eGFR was lower in the mGFR-cohort, where clinical decision making was based on mGFR, than in centers that only used eGFR. However, when taking time into account, we saw that both eGFR-cohort1 and eGFR-cohort2 accepted donors with lower pre-donation eGFR after 2009 compared to before 2009, resulting in disappearance of the differences in pre-donation eGFR. A reasonable explanation for this is that both centers accepted older donors after 2009 compared to before 2009, whereas in the mGFR-cohort there was no difference in eGFR and age before and after 2009. The increase in age and the consistency of BMI over time that we found in this study are consistent with previous results [19]. The introduction of a national living kidney donor guideline in the Netherlands in 2008, in

which age-adapted thresholds for pre-donation eGFR were introduced might have contributed to more uniformity in donor selection policies resulting in more similarity in recent donor characteristics [3]. Our findings are in line with a previous study by Gaillard et al., who concluded that mGFR is the most efficient method for living donor screening, but when not available, age-adapted thresholds for eGFR are also convenient [20]. Furthermore, we did not find differences in five-year post-donation eGFR, despite differences in pre-donation GFR assessment methods, which further supports the impression that routine use of mGFR does not have an effect on mean eGFR on population level.

While routinely using mGFR in donor screening does not have an effect on the total population characteristics, we did find that in half of the donors from the mGFR-cohort, mGFR was underestimated by eGFR $\geq$10 mL/min/1.73m$^2$ and in 20% of the donors even >20 mL/min/1.73m$^2$. Reasons for donor exclusion are mostly multifactorial and rarely solely based on insufficient kidney function. Still, kidney function plays a major role in the decision-making process of accepting a potential donor, and played an important role in the decision of 22% of the declined donors in the mGFR-cohort. If insufficient eGFR is a decisive factor in the decision to decline a potential donor, confirmative GFR assessment might be needed, especially in younger donors. This is supported by the finding that donors with the lowest 10% eGFR were younger in the mGFR-cohort (where mGFR was used) than in eGFR-cohort1 and eGFR-cohort2 (where eGFR was used). Measuring creatinine clearance (CrCl) from 24-hour urine samples might be an alternative. However, besides the sampling errors that could cause measurement inaccuracy, 24h CrCl tends to overestimate mGFR [21]. This overestimation increases in the lower ranges of GFR, possibly due to an increased tubular secretion of creatinine, causing an increased error in donors with marginal kidney function. For the majority of potential donors, 24h CrCl combined with eGFR will be sufficient to assess kidney function, because the mGFR will likely be in between those values. However for borderline cases with for example a too low eGFR and acceptable 24h CrCl, it is dangerous to assume that the 24h CrCl will be closer to the mGFR value than the eGFR value. In such cases additional mGFR testing would be useful.

The current guidelines do not clearly specify how GFR should be assessed before living kidney donation [2, 3, 6]. This study supports the concept that assessment of mGFR is not needed in every donor, but could be considered for a selected group of potential donors, for example young donors with an insufficient eGFR, consistent with previous results [20]. The previously developed online calculator from Huang et al., that calculates the probability to reach a specific pre-donation mGFR threshold based on pre-donation eGFR, age, sex and race, could be a supportive tool to distinguish between donors who could and who likely do not benefit from confirmatory mGFR testing [22]. In our study, only age was associated with an underestimation of mGFR by eGFR >20 mL/min/1.73m$^2$, and we did not identify other characteristics that led to underestimation of mGFR. Future studies should focus on more detailed characterization of donors in whom eGFR is inaccurate.

Strengths of this study include the extensive renal function measurements with $^{125}$I-Iothalamate in the mGFR-cohort. Furthermore, the comparisons were made in relatively large populations throughout the whole country with long-term follow-up. Also, consistent use of methods for kidney function determination in the centers limits confounding by indication. Yet, our study also has several limitations. First of all, the decision to accept a donor is multifactorial, and does not only rely on pre-donation GFR. Yet, we were able to identify 16 donors that were declined due to insufficient GFR and another 22 in whom GFR was one of multiple reasons for disapproval. Both estimated and measured GFR of these donors were lower than in the accepted donors. Data on declined donors in the other centers were not available. Lastly, the three populations mainly consisted of Caucasian donors. It is known that people of African

ancestry (i.e. African Americans, Black U.K. people) on average have higher muscle mass, possibly leading to larger underestimation of GFR by the creatinine-based equations [23]. However, because end-stage kidney disease is more prevalent among African and African American ethnicities [23], extra caution might be needed when accepting donors from these ancestries with lower pre-donation eGFR. Recently, it has been suggested to remove the racial correction factors in the eGFR equations, which led to more underestimation of GFR in black individuals in the general population as compared to white individuals. How these equations affect the applicability of the results of the current study (i.e. in a population with higher GFR than the general population) remains to be investigated.

In conclusion, this study shows that routinely measuring GFR using exogenous filtration markers did not lead to a detectable difference in the donor population compared to using eGFR. These results suggest that the routine use of mGFR does not seem to result in acceptance of donors with lower pre-donation eGFR on the population level, neither does it result in differences in five year post-donation eGFR. For the majority of potential donors eGFR and/or 24h CrCl may provide sufficient guidance. Future studies are needed to confirm our results and investigate whether a group could be identified (e.g. young donors) that might benefit from confirmatory mGFR testing.

## Supporting information

**S1 Fig. Distribution of pre-donation BMI before and after 2009 per center.** Differences between mean pre-donation BMI were tested using the independent sample T-test, P-values are shown in the figure.
(DOCX)

**S2 Fig. Distribution of five-year post-donation eGFR (CKD-EPI) per center.** Differences between mean five-year post-donation eGFR were tested using the independent sample T-test, P-values are shown in the figure. Five-year post-donation mGFR in the mGFR-cohort was added on the right in the figure.
(DOCX)

**S3 Fig. Distribution of five-year post-donation eGFR (CKD-EPI) before and after 2009 per center.** Differences between mean five-year post-donation eGFR were tested using the independent sample T-test, P-values are shown in the figure. Five-year post-donation mGFR before and after 2009 in the mGFR-cohort was added on the right in the figure.
(DOCX)

**S4 Fig. Distribution of pre-donation 24hCrCl before and after 2009 in the mGFR-cohort and eGFR-cohorteGFR-cohort2.** Differences between mean pre-donation 24hCrCl were tested using the independent sample T-test, P-values are shown in the figure. Pre-donation mGFR in the mGFR-cohort was added on the right in the figure.
(DOCX)

**S5 Fig. Distribution of pre-donation eGFR (CG) before and after 2009 per center.** Differences between mean pre-donation eGFR were tested using the independent sample T-test, P-values are shown in the figure. Pre-donation mGFR in the mGFR-cohort was added on the right in the figure.
(DOCX)

**S6 Fig. Distribution of pre-donation eGFR (MDRD) before and after 2009 per center.** Differences between mean pre-donation eGFR were tested using the independent sample T-test, P-values are shown in the figure. Pre-donation mGFR in the mGFR-cohort was added on the

right in the figure.
(DOCX)

**S1 Table. Pre- and five year post-donation bias between eGFR and mGFR$_{/BSA}$ in the mGFR-cohort.** Bias calculated as mGFR$_{/BSA}$−eGFR. Abbreviations: eGFR: Estimated glomerular filtration rate; mGFR$_{/BSA}$: Measured glomerular filtration rate corrected for BSA; BSA: Body surface area; IQR: Interquartile range.
(DOCX)

**S2 Table. Pre- and post-donation bias between CrCl and mGFR$_{/BSA}$ in the mGFR-cohort.** Bias calculated as CrCl−mGFR$_{/BSA}$. Abbreviations: CrCl: 24 hour creatinine clearance; mGFR$_{/BSA}$: Measured glomerular filtration rate corrected for BSA; BSA: Body surface area; SD: Standard deviation; IQR: Interquartile range.
(DOCX)

**S3 Table. Pre-donation characteristics of donors from the mGFR-cohort and eGFR-cohorteGFR-cohort2 with an underestimation of CrCl by eGFR ≥10 mL/min.** Binary variables presented as n (%), continuous variables presented as mean ±SD. Abbreviations: CKD-EPI: Chronic kidney disease epidemiology collaboration equation; CrCl: Creatinine clearance; mGFR: Measured GFR; BMI: Body mass index; BSA: Body surface area; SBP: Systolic blood pressure; DBP: Diastolic blood pressure; SD: Standard deviation.
(DOCX)

**S4 Table. Pre-donation characteristics of donors from the mGFR-cohort with an overestimation of mGFR$_{/BSA}$ by eGFR.** Binary variables presented as n (%), continuous variables presented as mean ±SD. Abbreviations: CKD-EPI: Chronic kidney disease epidemiology collaboration equation; CrCl: Creatinine clearance; mGFR: Measured GFR; BMI: Body mass index; BSA: Body surface area; SBP: Systolic blood pressure; DBP: Diastolic blood pressure; SD: Standard deviation.
(DOCX)

**S5 Table. Number of donors with eGFR above and under age-adapted threshold according to the Dutch Living Kidney Donor Guidelines.** Abbreviations: eGFR: Estimated glomerular filtration rate.
(DOCX)

**S1 File. GFR measurement mGFR-cohort.**
(DOCX)

## Acknowledgments

We would like to express our gratitude to Tessa Royaards for the excellent management of the Rotterdam database.

## Author Contributions

**Conceptualization:** Jessica van der Weijden, Marco van Londen, Joke I. Roodnat, Jacqueline van de Wetering, Heinrich Kloke, Ine M. M. Dooper, Martin H. De Borst, Stefan P. Berger.

**Data curation:** Jessica van der Weijden, Joke I. Roodnat, Jacqueline van de Wetering, Heinrich Kloke, Ine M. M. Dooper, Stephan J. L. Bakker, Gerjan Navis, Ilja M. Nolte, Martin H. De Borst, Stefan P. Berger.

**Formal analysis:** Jessica van der Weijden, Marco van Londen, Joke I. Roodnat, Marcia L. Kho, Heinrich Kloke, Ine M. M. Dooper, Ilja M. Nolte, Martin H. De Borst, Stefan P. Berger.

**Investigation:** Jessica van der Weijden, Marco van Londen, Joke I. Roodnat, Marcia L. Kho, Heinrich Kloke, Ine M. M. Dooper, Ilja M. Nolte, Martin H. De Borst, Stefan P. Berger.

**Methodology:** Jessica van der Weijden, Marco van Londen, Joke I. Roodnat, Jacqueline van de Wetering, Heinrich Kloke, Ine M. M. Dooper, Ilja M. Nolte, Martin H. De Borst, Stefan P. Berger.

**Project administration:** Jessica van der Weijden, Martin H. De Borst, Stefan P. Berger.

**Resources:** Jessica van der Weijden.

**Software:** Ilja M. Nolte.

**Supervision:** Marco van Londen, Joke I. Roodnat, Marcia L. Kho, Jacqueline van de Wetering, Heinrich Kloke, Ine M. M. Dooper, Stephan J. L. Bakker, Gerjan Navis, Ilja M. Nolte, Martin H. De Borst, Stefan P. Berger.

**Validation:** Joke I. Roodnat.

**Visualization:** Jessica van der Weijden, Marco van Londen.

**Writing – original draft:** Jessica van der Weijden, Marco van Londen, Joke I. Roodnat, Ine M. M. Dooper, Martin H. De Borst, Stefan P. Berger.

**Writing – review & editing:** Marco van Londen, Joke I. Roodnat, Marcia L. Kho, Jacqueline van de Wetering, Heinrich Kloke, Ine M. M. Dooper, Stephan J. L. Bakker, Gerjan Navis, Ilja M. Nolte, Martin H. De Borst, Stefan P. Berger.

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
