## [Decision Letter · Decision Letter 0]

3 Feb 2022

PONE-D-22-00925Impact of measured versus estimated glomerular filtration rate-based screening on living kidney donor characteristics: A multicenter cohort study.PLOS ONE

Dear Dr. van der Weijden,

Thank you for submitting your manuscript to PLOS ONE. After careful consideration, we feel that it has merit but does not fully meet PLOS ONE’s publication criteria as it currently stands. Therefore, we invite you to submit a revised version of the manuscript that addresses the points raised during the review process.

We look forward to receiving your revised manuscript.

Kind regards,

Frank JMF Dor, M.D., Ph.D., FEBS, FRCS

Academic Editor

PLOS ONE

Journal Requirements:

2. We note that your study involved tissue/organ transplantation. Please provide the following information regarding tissue/organ donors for transplantation cases analyzed in your study if you have not already done so.

a. Please provide the source(s) of the transplanted tissue/organs used in the study, including the institution name and a non-identifying description of the donor(s).

b. Please state in your response letter and ethics statement whether the transplant cases for this study involved any vulnerable populations; for example, tissue/organs from prisoners, subjects with reduced mental capacity due to illness or age, or minors.

- If a vulnerable population was used, please describe the population, justify the decision to use tissue/organ donations from this group, and clearly describe what measures were taken in the informed consent procedure to assure protection of the vulnerable group and avoid coercion. 

- If a vulnerable population was not used, please state in your ethics statement, “None of the transplant donors was from a vulnerable population and all donors or next of kin provided written informed consent that was freely given.”

c. In the Methods, please provide detailed information about the procedure by which informed consent was obtained from organ/tissue donors or their next of kin. In addition, please provide a blank example of the form used to obtain consent from donors, and an English translation if the original is in a different language.

d. Please indicate whether the donors were previously registered as organ donors. If tissues/organs were obtained from deceased donors or cadavers, please provide details as to the donors’ cause(s) of death.

e. Please ensure the participant recruitment dates and the period during which transplant procedures were done (as month and year) are provided. 

f. Please discuss whether medical costs were covered or other cash payments were provided to the family of the donor. If so, please specify the value of this support (in local currency and equivalent to U.S. dollars).

Additional Editor Comments:

Many thanks for submitting your work to Plos One. Your MS has now been reviewed by 2 experts in the field and they have come up with quite a few concerns regarding methodology with regards to design, analysis and data interpretation. They will need to be addressed in a point-by-point fashion and revisions should be made accordingly based on the criticisms. Please also involve a native English speaker to edit the MS. This is not guarantee for acceptance as the revised MS will undergo thorough peer review again. If you feel you can't address the criticisms satisfactorily, please let us know. Im personally looking forward to the discussion with the reviewers, and hope to see your rebuttal and revised MS in due course. Best wishes!

Reviewers' comments:

Reviewer's Responses to Questions

**Comments to the Author**

1. Is the manuscript technically sound, and do the data support the conclusions?

Reviewer #1: No

Reviewer #2: Yes

2. Has the statistical analysis been performed appropriately and rigorously? 

Reviewer #1: No

Reviewer #2: Yes

3. Have the authors made all data underlying the findings in their manuscript fully available?

Reviewer #1: Yes

Reviewer #2: Yes

4. Is the manuscript presented in an intelligible fashion and written in standard English?

Reviewer #1: Yes

Reviewer #2: Yes

5. Review Comments to the Author

Reviewer #1: Thank you for asking me to review this paper. I’m afraid I don’t think the study as presented is well designed to answer the questions posed in the title and introduction, as no assessment of the e/mGFR result on decision making regarding donor acceptance is made. e/mGFR results for screened but non-proceeding donors are not presented. The cohorts compared are not directly comparable (e.g., donors from different time periods from different centres with different decision makers are being compared, no attempt is made to undertake or present an analysis adjusted for potential confounders). I have provided further detailed comments below.

Title

1. 'Impact of measured versus estimated glomerular filtration rate-based screening on living kidney donor characteristics: A multicenter cohort study.' - The study is not a multicenter single cohort study as implied. It appears to be a study of multiple cohorts?

Abstract

2. Line 23: ‘Most transplant centers use estimated glomerular filtration rate (eGFR) for evaluation of potential living kidney donors.’ – This statement isn’t true for centers outside the Netherlands e.g., in the USA and UK kidney function is measured prior to donation. Suggest edit this sentence to read ‘Most transplant centers in the Netherlands’

3. Line 24: ‘eGFR underestimates GFR’ – This statement isn’t true for all. eGFR is likely to overestimate kidney function in those with low BMI, and individuals from certain ethnic groups e.g., Chinese, East Asian. Suggest reword as ‘eGFR can underestimate GFR’ or ‘eGFR often underestimates GFR in healthy donors’.

4. Line 27: I think the word ‘acceptation’ should be ‘acceptance’.

5. Abstract results section: you need to state what results are being presented. You state that ‘Donor age was similar among the cohorts mGFR-cohort 53+/- 10 years’ but you don’t state whether this is mean age with standard deviation, or median age with IQR presented. The same is true for the GFR results – are you presenting means? If so state this and state what the +/- relates to, presumably standard deviation?

6. Line 38: Please can you give the specific p value, not ‘p<0.05’. It is difficult to interpret whether there is strong evidence to support a difference e.g., p=0.001 vs weak evidence to support a difference e.g., p=0.048. There is nothing special about a p value of 0.05 that makes something significant or not, it’s an arbitrary threshold. See: https://www.tandfonline.com/doi/full/10.1080/00031305.2019.1583913 This point applies to the methods in which it is stated that a p value <0.05 is 'statistically significant'.

7. Lines 44 – 46: ‘we did not show that the routine use of mGFR in donor screening leads to inclusion of donors with a lower pre-donation eGFR’ – This finding hasn’t been presented in the results section of the abstract and should be if it is to be referenced here. Suggest adding proportion of those not donating due to inadequate e/mGFR added to each cohort?

8. Line 47: ‘but young donors with insufficient eGFR might benefit from confirmatory mGFR testing’ – There’s no evidence presented in the abstract to justify this statement. Suggest adding to results or removing the statement.

Introduction

9. Lines 51-53: ‘The main goal of living kidney donor evaluation is to assess whether a donor will retain sufficient kidney function after donation for a life-long safe kidney function.’ – This sentence isn’t quite true. The main goal is not just to assess whether a donor will be left with adequate kidney function after donation, but also to determine if they will survive the nephrectomy operation (people may have more than adequate kidney function but be unsuitable for other medical reasons) and to determine if they are psychologically fit to donate.

10. Line 57: ‘The latter is expensive and laborious and therefore much less widespread in use’ is not true. Iohexol is cheap and iohexol mGFR is easy to do. Please also state that you are still only talking about the Netherlands, in the UK and USA kidney function is measured prior to donation.

11. Line 69: ‘higher ranges of GFR, true GFR is underestimated’ please edit this to either state ‘in most White populations’ or ‘most White and Black populations’– I think all the references provided to support this present evidence from largely White European and White American populations, which aren’t the Global majority.

12. Line 75: ‘long-term safety was studied by comparing kidney function five years after donation’ – 5 years after donation is not long-term. In living kidney donation the 20 year follow up data is considered medium term so 5 years is really short term.

Methods

13. Lines 89-90 – please state whether the mGFR cohort have also given written informed consent.

14. Lines 98 and 116 and 126: Why are cohorts for different time periods being compared? As indicated the time/year of assessment may have influenced acceptance of the kidney donors. The cohorts compared should be for the same time periods. This is a major issue – like is not being compared with like.

15. Line 148: ‘mGFR>10’ later in the results you state ‘mGFR ≥10’ – please correct whichever is wrong.

16. Line 148: ‘mGFR>20’ do you mean ‘mGFR≥20’? If so you need to change this in the results as well (e.g., line 173)

17. Lines 152-154: ‘Because reason of exclusion from donation was mostly multifactorial and rarely solely dependent GFR, we did not analyze the number of donors excluded based on kidney function per center.’ – It seemed that the point of the study was to determine if people with low eGFR were being inappropriately prevented from donating because of their inadequate kidney function, therefore this question of reason for exclusion is critical. Otherwise there is little point studying this in donors specifically – you could have simply compared eGFR and mGFR in the same healthy population. The whole aim of the study is to determine the impact of measured versus estimated GFR based screening on living kidney donor acceptance – without presenting data regarding this question the study adds little to existing knowledge re. the performance of m and eGFR measurements.

Results

18. I’m surprised not to see any Bland-Altman plots comparing m and eGFR measurements?

19. Table 1 lacks units in the first column, and an indication of what is being presented outside and within brackets– these need to be added e.g., ‘CKD-EPI’ should be replaced with ‘mean CKD-EPI eGFR +/- standard deviation (ml/min/1.7m2)’; ‘Age’ should be replaced with ‘mean/median age in years +/- SD/IQR’ ‘Sex’ should have ‘Female sex, n (%)’ in the table. All GFRs needs units, as does weight, length, BMI, BSA, SBP, DBP, serum creat etc.

The same is true for Table 2 which needs indications as to whether the results are means, medians etc.

19. Table 2: Why are no tests for difference presented to compare these groups? This isn’t an RCT so it is justifiable to test how comparable the different cohorts are – differences between the groups is important to identify. If they are not comparable then to compare e and mGFRs between the groups without adjustment is not meaningful.

20. I also expected data to presented on those SCREENED for donation, given the title of the paper and the question of the e/mGFR on acceptance of the donor, yet I can’t see any data on those screened who didn’t donate.

21. The analysis of differences in eGFR over time doesn’t address the original question of the paper, and without knowing the impact on the donor (i.e. did donors post 2009 with lower eGFRs get accepted more than before 2009) it adds little to the analysis.

22. Lines 235 and 236: What are the p values attached to? What comparison is being made? It appears to be that you are comparing the eGFR cohorts to the mGFR cohort?

23. Table 3: Table 3 is just presenting data on the same individuals as in Table 2 – so there is no point reproducing variables that won’t change e.g. sex, race.

Discussion

24. Line 271: ‘This study aimed to investigate the effect of using mGFR versus eGFR on donor selection and long-term post-donation kidney function’ – I am afraid the study has not achieved its aims. The analysis presented doesn’t tell me anything about the impact of the e/mGFR on SELECTION and 5 years follow up data is not long-term.

25. I’m afraid a fully adjusted analysis is required to conclude that ‘mGFR based screening allows acceptance of donors with lower pre-donation eGFR than a center that only uses eGFR’.

26. Lines 337-339: ‘It is known that people of African ancestry on average have higher muscle mass, possibly leading to larger underestimation of GFR by the creatinine-based equations.’ – This statement is not correct. This has only really been well described in African ‘migrant’ populations e.g., African Americans, Black UK, and not been well described in African populations in Africa.

Reviewer #2: thank you for asking me to review this paper which confirms that in most cases measuring mGFR did not lead to a detectable difference in donor population compared to using eGFR.

the paper is useful as mGFR is time consuming and expensive and therefore demonstrating that in most cases the additional effort is nor justified.

I wondered if you might give the manuscript to a native English speaker as whilst it is very well written there are a few anomalies in the use of English

the sentence in lines 306-7 has become scrambled

the authors comment on possible differences in estimation of GFR in different racial groups - the authors may want to briefly review the most recent American and British guidance on this which suggests removing correction factors due to race as this is now considered to be an over simplistic classification

the important point about possible continued utility in younger donors is well made

I wondered if the paper might emphasise more the long term safety data comparing kidney function 5 years after donation an not demonstrating any difference depending on original method of GFR measurement for donation

6. PLOS authors have the option to publish the peer review history of their article (what does this mean?). If published, this will include your full peer review and any attached files.

Reviewer #1: No

Reviewer #2: No

---

## [Decision Letter · Decision Letter 1]

12 Jun 2022

PONE-D-22-00925R1Impact of measured versus estimated glomerular filtration rate-based screening on living kidney donor characteristics: A study of multiple cohorts.PLOS ONE

Dear Dr. van der Weijden,

Thank you for submitting your manuscript to PLOS ONE. After careful consideration, we feel that it has merit but does not fully meet PLOS ONE’s publication criteria as it currently stands. Therefore, we invite you to submit a revised version of the manuscript that addresses the points raised during the review process.

ACADEMIC EDITOR:Apologies for the delay.conditionally accepted, pending minor revision.

We look forward to receiving your revised manuscript.

Kind regards,

Frank JMF Dor, M.D., Ph.D., FEBS, FRCS

Academic Editor

PLOS ONE

Journal Requirements:

Reviewers' comments:

Reviewer's Responses to Questions

**Comments to the Author**

1. If the authors have adequately addressed your comments raised in a previous round of review and you feel that this manuscript is now acceptable for publication, you may indicate that here to bypass the “Comments to the Author” section, enter your conflict of interest statement in the “Confidential to Editor” section, and submit your "Accept" recommendation.

Reviewer #1: All comments have been addressed

2. Is the manuscript technically sound, and do the data support the conclusions?

Reviewer #1: Yes

3. Has the statistical analysis been performed appropriately and rigorously? 

Reviewer #1: Yes

4. Have the authors made all data underlying the findings in their manuscript fully available?

Reviewer #1: No

5. Is the manuscript presented in an intelligible fashion and written in standard English?

Reviewer #1: Yes

6. Review Comments to the Author

Reviewer #1: Thank you for addressing my concerns, and making significant changes to the analysis and manuscript. The paper is much improved, and the more cautious conclusions more appropriate given the limitations of the study.

Table 2 and Table 3 'P vs. mGFRcohort1' should this be 'vs mGFR-cohort' as there's only one.

Table 2 and Table 4: Length should be height

7. PLOS authors have the option to publish the peer review history of their article (what does this mean?). If published, this will include your full peer review and any attached files.

Reviewer #1: No

---

## [Author Response · Author response to Decision Letter 1]

20 Jun 2022

Reviewers' comments:

Have the authors made all data underlying the findings in their manuscript fully available?

Reviewer #1: No

Author reply: Thank you for reviewing our revised manuscript. We now included a data availability statement at the end of the manuscript.

Review Comments to the Author

Reviewer #1: Thank you for addressing my concerns, and making significant changes to the analysis and manuscript. The paper is much improved, and the more cautious conclusions more appropriate given the limitations of the study.

Table 2 and Table 3 'P vs. mGFRcohort1' should this be 'vs mGFR-cohort' as there's only one.

Table 2 and Table 4: Length should be height

Author reply: We changed “mGFR-cohort1” to “mGFR-cohort” and “Length” to “Height” throughout the manuscript and supplementary data accordingly.

---

## [Editor Report · Decision Letter 2]

21 Jun 2022

Impact of measured versus estimated glomerular filtration rate-based screening on living kidney donor characteristics: A study of multiple cohorts.

PONE-D-22-00925R2

Dear Dr. van der Weijden,

We’re pleased to inform you that your manuscript has been judged scientifically suitable for publication and will be formally accepted for publication once it meets all outstanding technical requirements.

Kind regards,

Frank JMF Dor, M.D., Ph.D., FEBS, FRCS

Academic Editor

PLOS ONE
---

## [Editor Report · Acceptance letter]

27 Jun 2022

PONE-D-22-00925R2 

Impact of measured versus estimated glomerular filtration rate-based screening on living kidney donor characteristics: A study of multiple cohorts. 

Dear Dr. van der Weijden:

I'm pleased to inform you that your manuscript has been deemed suitable for publication in PLOS ONE. Congratulations! Your manuscript is now with our production department. 

Kind regards, 

on behalf of

Dr. Frank JMF Dor 

Academic Editor

PLOS ONE